# A New Diversity Panel for Winter Rapeseed (Brassica *napus*, L.) Genome-Wide Association Studies

**David P. Horvath [1],\***, **Michael Stamm [2]**, **Zahirul I. Talukder [3]**, **Jason Fiedler [1]**,
**Aidan P. Horvath [4]**, **Gregor A. Horvath [5]**, **Wun S. Chao [1]** and **James V. Anderson [1]**

[1] Edward T. Schafer Agricultural Research Center, 1616 Albrecht Blvd, Fargo, ND 58102, USA;
jason.fiedler@usda.gov (J.F.); wun.chao@usda.gov (W.S.C.); james.v.anderson@usda.gov (J.V.A.)

[2] Department of Agronomy, 3702 Throckmorton Plant Sciences Center, Kansas State University,
Manhattan, KS 66506, USA; mjstamm@ksu.edu

[3] Department of Plant Sciences, 166 Loftsgard Hall, North Dakota State University, Fargo, ND 58105, USA;
zahirul.talukder@ndsu.edu

[4] Department of Neurobiology, University of Michigan, Ann Arbor, MI 48109, USA; horvathap@gmail.com

[5] Human Development and Family Studies, Perdue University, Purdue Mall, West Lafayette, IN 47907, USA;
gregor.a.horvath@gmail.com

**\*** Correspondence: david.horvath@usda.gov

**Abstract:** A diverse population (429 member) of canola (Brassica *napus*, L.) consisting primarily of winter biotypes was assembled and used in genome-wide association studies. Genotype by sequencing analysis of the population identified and mapped 290,972 high-quality markers ranging from 18.5 to 82.4% missing markers per line and an average of 36.8%. After interpolation, 251,575 high-quality markers remained. After filtering for markers with low minor allele counts (count > 5), we were left with 190,375 markers. The average distance between these markers is 4463 bases with a median of 69 and a range from 1 to 281,248 bases. The heterozygosity among the imputed population ranges from 0.9 to 11.0% with an average of 5.4%. The filtered and imputed dataset was used to determine population structure and kinship, which indicated that the population had minimal structure with the best K value of 2–3. These results also indicated that the majority of the population has substantial sequence from a single population with sub-clusters of, and admixtures with, a very small number of other populations. Analysis of chromosomal linkage disequilibrium decay ranged from ~7 Kb for chromosome A01 to ~68 Kb for chromosome C01. Local linkage decay rates determined for all 500 kb windows with a 10kb sliding step indicated a wide range of linkage disequilibrium decay rates, indicating numerous crossover hotspots within this population, and provide a resource for determining the likely limits of linkage disequilibrium from any given marker in which to identify candidate genes. This population and the resources provided here should serve as helpful tools for investigating genetics in winter canola.

**Keywords:** canola; diversity panel; population structure; genome-wide association study; linkage decay

## 1. Introduction

Winter rapeseed is an economically valuable, cool season oilseed crop grown in the United States, ranging from Idaho to the central Mid-Atlantic states. It has also gained attention as a potential winter cover crop, with traits suitable for developing dual-cropping systems, since it generally outperforms spring biotypes in yield and matures early enough to allow for harvest before planting of some varieties of corn, soybean, and sunflower [1]. As a cover crop, winter rapeseed could also provide ecosystem

benefits such as a weed control, reduced soil erosion and nutrient leaching, and additional sources of early season pollinator habitat [2]. However, there are a number of traits that breeders would need to optimize in order to exploit the full potential of this crop. Many of these traits—such as flowering time and freezing tolerance—are multigenic and pleiotropic in nature and, thus, are time consuming to optimize in elite germplasm [3].

Genome-wide association studies (GWAS) are a commonly used method to assess and mark genomic regions that influence polygenic traits [4]. However, GWAS requires a diverse population of a given species to utilize the genetic diversity that has built up through evolutionary time [4]. This genetic diversity provides the large numbers of polymorphic genetic markers needed to associate specific loci to the phenotypic diversity for any given trait in that population. Because the scale of diversity within the populations often results in high crossover rates between markers and the various mutations that might impact a trait of interest, GWAS requires exceptionally high marker density when the linkage disequilibrium decay rate is high [5].

There have been several populations of canola that have been previously genotyped for use in GWAS studies [6–13]. The majority of these diversity panels were a mix of winter, semi-winter, and spring lines, and none had more than 200 winter varieties included in the population. Moreover, in most cases, the number of genotyped single nucleotide polymorphisms (SNPs) was fewer than 100K. Finally, only one of these genotyped panels is available in the US. These observations point towards the need for a readily available, high-quality, and well-characterized diversity panel for winter rapeseed.

Genotype by sequencing (GBS) is an effective means to generate the high density of markers needed to identify genetic loci associated with a given trait of interest using a GWAS-based approach [5]. However, a large population can have a fair number of individuals with numerous mutations that are only unique to that individual. Because these mutations are not likely to be highly informative, GBS data are often filtered to remove individuals within the population that produced poor sequence and thus have a high percentage of missing markers, and also to filter out loci where only a few individuals within the population have a given variant [5]. Once a given population is genotyped and an adequate number of high-density markers are identified in an arbitrary percentage of the population (usually greater than 5 or 10%), a statistical method, known as imputation, can be employed to predict the genotype of missing markers in a given individual based on the prevalence of non-missing markers at that location from obviously related individuals [5]. The resulting file (usually referred to as an imputed VCF file) can then be used to look for markers that are in linkage disequilibrium (linked) to genes controlling any trait of interest that might be variable within the population. However, assessing the effectiveness of the process should be done to provide evidence that it was effective on any given population.

Here, we describe a GBS mapped population of primarily winter rapeseed biotypes with higher numbers of high-quality SNPs and larger population size than has been previously reported for canola. The verified imputed VCF file and other resources are made available for use, and the seeds for each line will be made available on request. We also describe a novel method to identify local linkage decay rates that has proven exceptionally useful for identifying candidate genes for freezing tolerance and deacclimation processes in GWAS studies using this population [14].

## 2. Materials and Methods

### 2.1. Plant Material

A diversity panel of 433 primarily winter rapeseed populations was assembled from a breeding population of winter rapeseed at Kansas State University and represents winter rapeseed biotypes primarily from North America and Europe. Although all of the lines were presumably winter biotypes, 19 lines may have been mis-classified based on the observations that they flowered without the need for vernalization, and 20 lines were known spring types. These lines should be considered for any projects to map flowering time genes. Of the primarily winter rapeseed lines, 238 had been advanced

to the F9 generation via selfing prior to DNA extraction. The remaining lines had been advanced to between the F5 and F8 generation. A list of common cultivar names associated with the associated Agricultural Research Service (ARS) numbers and preliminary phenotypic traits, including two years of data for flowering time, are available in Supplemental File S1.

*2.2. Genotype by Sequencing and Polymorphism Filtering and Assessment*

DNA was isolated from a single plant from each line using the DNeasy 96 Plant Kit (Qiagen, Hilden, Germany) with modifications. Samples were lysed with FastPrep-24™ 5G Homogenizer (MP Biomedicals, Irvine, CA, USA). The settings for the instrument were: two pulses at 6.0 m/s for 20 s each with a 10 s pause between pulses. Lysis tubes were spun at $16,000 \times g$ in a microcentrifuge. After the addition of Buffer P3, samples were incubated on ice for 10 min. S-Blocks were modified for use in a Sorval SH3000 swing-bucket rotor. Samples were spun at $3827 \times g$ (4300 rpm). To bind the DNA, 600 μL of sample was applied twice to the DNeasy 96 plate (because of the reduced volume of the modified S-Blocks). Membrane was washed twice with 400 μL of Buffer AW2 (because of the reduced volume of the modified S-Blocks), each with a 5 min spin. Membrane was dried with a final 10 min spin. DNA was eluted off the membrane twice with 100 μL Buffer AE. Buffer was added to the membrane, incubated for 1 min at room temperature, then spun for 2 min at $3827 \times g$. The elutions were pooled before quantitation. DNA was quantitated using Quant-iT™ PicoGreen® dsDNA Kit (Life Technologies, P7589) and a fluorescent plate reader (Varioskan LUX, Thermo Scientific, Waltham, MA, USA) with SkanIt analysis software (Thermo Scientific, Waltham, MA, USA). Quality of the extracted DNA was determined by visually observing selected samples run on a gel alongside the same samples cut with restriction enzyme. For each 20 μL reaction: 6 μL (~300 ng) DNA, 4 U Hind III (Roche, Basel, Switzerland), 2 μL reaction buffer, and H20. Reactions were incubated for 2 h at 37 °C. Then, 2.2 μL loading buffer (Invitrogen, Carlsbad, CA, USA,) was added to each reaction before loading on a 1% agarose gel with size standard. Then, 2 μL (~100 ng) undigested DNA was diluted to 20 μL, combined with loading buffer and loaded alongside the corresponding digested sample. Gel was run at 4 V/cm for 1 h and imaged using BioRad gel imager and software (Hercules, CA, USA).

The resulting DNA was sent to Cornell University and the enzyme ApeKI was used to select fragments for building the GBS libraries. Sequencing was done on 5 lanes of Illumina 2500 with 150 base paired end reads as output. Over 3 billion reads were generated with 46% containing readable barcodes and subsequently over 8 million tags present at least three times were identified in the merged files. The resulting sequences were trimmed to remove contaminating and low-quality sequences and then clustered and mapped to the canola reference genome (Brassica_napus_v4.1.chromosomes.fa) using the TASSEL5 pipeline [15]. Over 5 million of these aligned uniquely to the canola genome. The resulting single nucleotide and indels variants were filtered as described [15] and based on the number of missing alleles in individuals and or at specific sites in the genome. This selection reduced the population size to 429 individuals with at least 15% called variants and 290,972 markers that were called in at least 10% of the population. This file is available as Supplemental File S2. The resulting high-quality marker collection was subjected to imputation using the program NGSEP_Impute_VCF_2.1.4 [16] in the CyVerse discovery environment [17], yielding 251,576 markers. The imputed vcf file can be found in Supplemental File S3. This file was further filtered for minor allele counts greater than 5 using Tassel5. The filtered imputed file is available as Supplemental File S4. Basic information on observed and expected allelic frequencies was determined using the program NGSEP Diversity Statistics 2.1.4 in the CyVerse discovery environment and can be found in Supplemental File S5.

*2.3. Imputation Assessment*

The imputation program, NGSEP_Impute_VCF_2.1.4 [16], was tested for effectiveness by using a script to artificially delete a random proportion of existing, non-missing genotype values. The sample vcf file was processed using Python's pandas library. Once loaded into a pandas data-frame, a custom python script was used to generate a list of random, non-missing data points, along with their

corresponding IDs and line IDs to store their coordinates within the data structure. These data points were then replaced with the characters used to indicate missing data ('./.' in this case). For our test, we replaced 0.0002%, approximately 13,000 of the non-missing genotype values in the file "all.taxamerged.filtered.recode.vcf.gz". A list of replaced values was saved as a csv file.

The imputation program was then run on the file with the 13,000 replacements, yielding a new file where the replaced values were imputed. Finally, a custom python script was used to compare each replaced genotype value from the list of replaced values against its corresponding data point in the imputed file (Supplemental File S3). We then calculated the percentage of correctly imputed data points overall and for each genotype.

## 2.4. Population Analysis

The program TASSEL5 was used to produce a kinship matrix (Supplemental File S6) based on the imputed vcf file (Supplemental File S4) using the default option "centered IBS". Model-based Bayesian clustering to infer the sub-population structure was performed with STRUCTURE (v 2.3.4) [18]. To speed up computation, genotype calls with a genotype quality (QG) score less than 50 were removed and loci missing more than 5% of the genotypes were removed and thinned to 1/10 kb or 1/50 kb, thus resulting in two genotyping datasets. The remaining 25,665 or 11,386 single-nucleotide polymorphisms (SNPs), for the 10 or 50 kb datasets, respectively, were independently analyzed with the default admixture model (K: 2-12; 9 replications; burn-in 20,000; simulation runs 80,000). CLUMPP [19] was used to align independent runs and allow comparative cluster allocations at each K value. An individual was assigned to the cluster that the largest fraction of its genotypes belonged to, with designation of admixtures set at <50%. Individual lines associated with each bar in the cluster analysis histogram can be found in Supplemental File S7. Due to the nature of the breeding population, the ad hoc methods described [20] were not useful in judging the number of optimal Bayesian clusters. Instead, we examined the change in the average Ln PR (X|K) values between K and K + 1 and analysis of each individual's proportional cluster allocation was also useful in determining K values exhibiting uninformative admixture. Nucleotide diversity ($\pi$) with a window size of 50 kb to approximate the average linkage decay rate with a 10 kb step and Tajima's D with a a window size also of 50 kb were calculated using subroutines in the program VCFtools [21]. These data can be found in Supplemental File S8. An unrooted neighbor-joining tree was constructed using Tassel5 and can be found in Supplemental File S9.

## 2.5. Linkage Disequilibrium Decay Analysis

Pairwise linkage disequilibrium (LD) was calculated in TASSEL5 [22] for all SNPs on each chromosome as the squared allele–frequency correlations (r2) following [23] (see Supplemental File S10 for the distance matrix). The decay of linkage disequilibrium (LD) with physical distance was estimated using a nonlinear regression analysis of linkage disequilibrium (LD) between SNP marker loci (r2) versus the genetic distance between SNP sites in base pairs [24]. Under the drift–recombination equilibrium, the expected value of r2 is $E(r^2) = 1/(1 + C)$, where C = 4Nc, N is the effective population size, and c is the recombination fraction between SNPs [25]. Under the assumption of a low level of mutation and an adjustment of sample size, the expectation becomes [26]:

$$E(r^2) = [(10 + C)/((2 + C)(11 + C))][1 + ((3 + C)(12 + 12C + C^2))/(n(2 + C)(11 + C))]$$

The data were pooled for the canola A and C genome and fit this equation into a nonlinear regression model using the statistical package R version 3.4.3 (R Development Core Team 2017). These data are available in Supplemental File S11. We also measured the rate of LD decay individually for all nineteen canola chromosomes.

Local linkage decay rates were determined using a similar method with a custom R script to determine the LD decay rate in all 500 kb increments across all chromosomes using a 10 kb step

(Supplemental File S12 provides the R script and Supplemental File S13 provides a table showing LD decay values for each window).

## 2.6. Phenotypic Analysis of Agronomic Traits of Interest

In order to assess the possibility of genetic variation for traits of agronomic interest in these populations, the diversity panel populations were planted in the fall of 2015 near Manhattan, KS and were arranged in a one replicate test with repeated checks in both growing seasons. The experimental setup had three row plots with a plot size of 3 m long by 0.76 m wide. Seeding rate was equal to common seeding rate for winter oilseed rape in the southern Great Plains 4.3 kg ha$^{-1}$ or approximately 140 seeds m$^{-2}$ under dryland conditions with no irrigation. Soil types in 2015–2016 were Ivan and Kennebec silt loam (fine-silty, mixed, superactive, mesic Cumulic hapludolls) and in 2017–2018 were Rossville silt loam (fine-silty, mixed, superactive mesic Cumulic Hapludolls).

The following phenotypic traits were measured: stand establishment (rated on a 0 to 10 scale with 0 = no stand and 10 = complete stand), plant vigor (rated on a 1 to 5 scale with 1 = least vigorous to 5 = most vigorous), and stem elongation (rated on a 1 to 5 scale with 1 = excessive and 5 = no stem elongation). Phenotypic traits measured in the spring included: rosette growth habit (rated on a 1 to 3 scale with 1 = upright, 2 = intermediate, and 3 = prostrate); winter survival (rated as the percentage of fall stand surviving the winter); days to flowering date after 1 January when 50% of plants within a plot had achieved one open flower); and plant height (measured in inches). A second measure for days to flowering was similarly measured in 2017. Above normal spring temperatures in 2016 resulted in much earlier flowering times. Dry conditions and below normal spring temperatures in 2018 resulted in later flowering times. Meteorological data for the field location in both years can be found in Supplemental File S14.

## 3. Results

### 3.1. SNP Identification and Filtering

GBS identified ~300,000 SNPs among the 433 winter rapeseed lines that were sequenced. After filtering and imputation, we were left with 429 lines with 251,575 single nucleotide polymorphisms (SNPs) mapped to the reference canola genome. An analysis of the imputation program indicated that 90.67% of the total data points were correctly imputed. We found that the imputation program worked well on homozygous genotypes, imputing 99.81% of major alleles and 85.07% of minor alleles. The imputation program was unable to correctly impute any heterozygous genotypes. However, these were rare as *B. napus* is a primarily self-pollinating species (see below). This imputed vcf file is available as Supplemental File S2. Analysis of the imputed SNPs (Supplemental File S3) indicated that 97,666 loci (~38%) had a minor allele frequency >0.05 but that 190,375 had minor allele counts >5.0. The average expected heterozygosity for all alleles was 13.7%, which was greater than the average observed heterozygosity (6.6%), as might be predicted for a primarily inbreeding species. Seeds from these lines will be made available upon request. The distances between SNPs range from 1 base to 281,248 bases, with 4462.8 and 69 bases for the average and median distances, respectively. Marker density was not even, and clustering of markers can be observed in regions with small and large distances between markers (see Figure 1 for an example). Analysis of LD for a given chromosome is shown in Figure 2, and two range from 7 kb to ~65 kb. Regression analysis of the LD indicated that the distance between SNPs decreased sharply with R2 values greater than 0.2, with a value of 11.35 kb for the A genome and 33.88 kb for the C genome (Supplemental File S11). Thus, the marker density is sufficiently high enough to expect viable associations for most loci. More localized LD indicated a range of LD decay intervals from 1 to 491,136 bases for any given 500 kb window (Supplemental File S13). An example for the LD decay along chromosome A01 is shown (Figure 3).

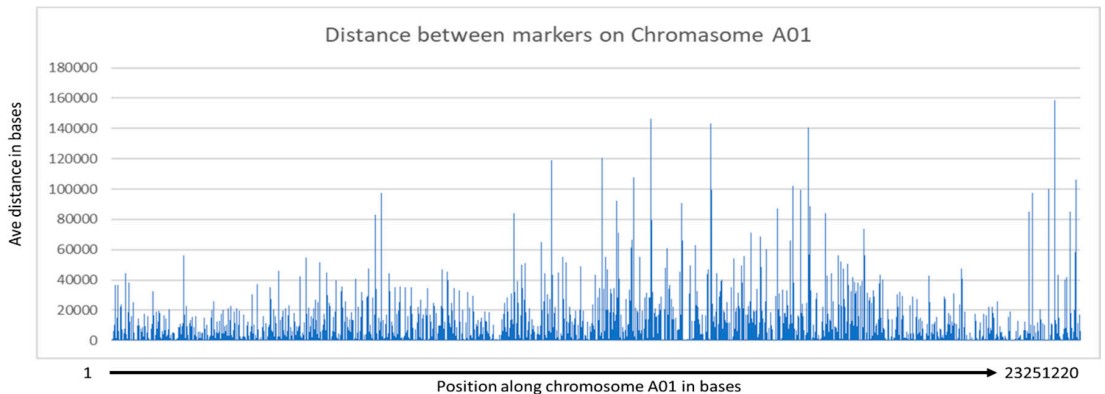

**Figure 1.** Distance between marker along chromosome A01.

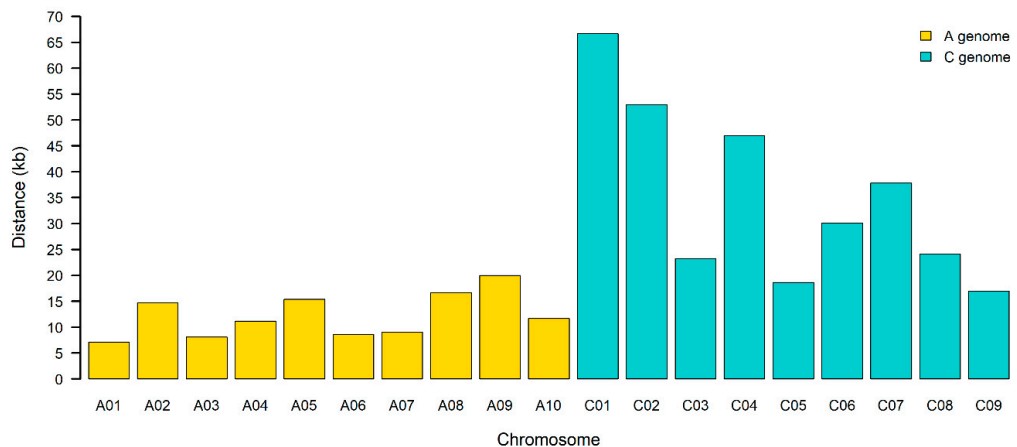

**Figure 2.** Linkage disequilibrium decay rate for each canola chromosome.

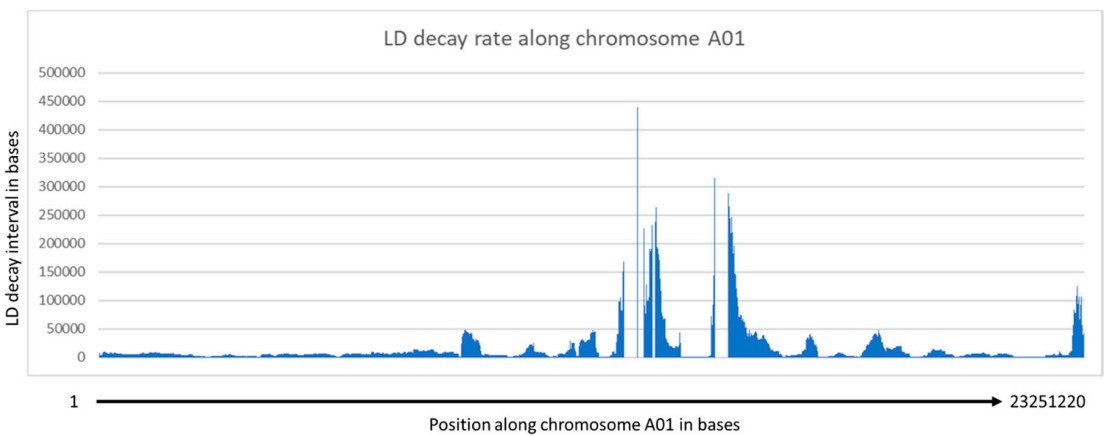

**Figure 3.** LD decay rate in bases along chromosome A01. Regions of highly repressed recombination among members of the population are observed as peaks.

Nucleotide diversity and Tajima's D values were calculated along the chromosomes. There was relatively little difference between the average values between the A and C genome ($4.89 \times 10^{-5}$ and $3.82 \times 10^{-5}$ for $\pi$, and 0.51 and 0.4 for Tajima's D values, respectively) (Supplemental File S8). There were localized troughs in $\pi$ that corresponded to negative localized values for Tajima's D in the same region. These may be indicative of regions undergoing some selection within the entire population. It should be noted, though, that this population is part of an active winter *B. napus* breeding program and thus some selection for agronomic and domestication traits might be expected.

## 3.2. Population Analysis

The population structure of our winter rapeseed lines was found to have significant genetic variability with 2–4 major clusters depending on the type of analysis conducted (Figures 4 and 5). The distance between individuals within any given cluster is generally consistent, as are the distances between clusters. Additionally, the neighbor-joining phylogenetic tree indicates a similar number of major branches (Supplemental File S9). Thus, we conclude that there is considerable genetic diversity in this population but that there are genetic relationships among the members of this diversity panel. Thus, predicted models that include kinship relationships and population structure will usually provide the most accurate inferences. This was proven to be the case, not only with this study but also with a parallel study on freezing tolerance following deacclimation using this same population [14].

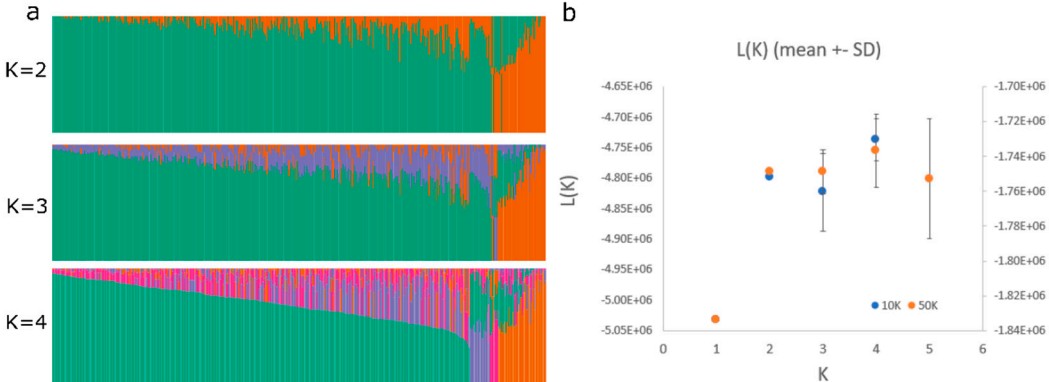

**Figure 4.** Population structure of the diversity panel population as determined using STRUCTURE for 5 different K values with a sub-population of markers representing one marker every 50,000 bases. Admixture set at <40%. (**a**). The K vs. Ln Likelihood plot is also shown for two different marker sets with one marker every 10,000 bases and every 50,000 bases, the most likely K value being 2−4 with the 1 marker/50,000-base dataset (**b**).

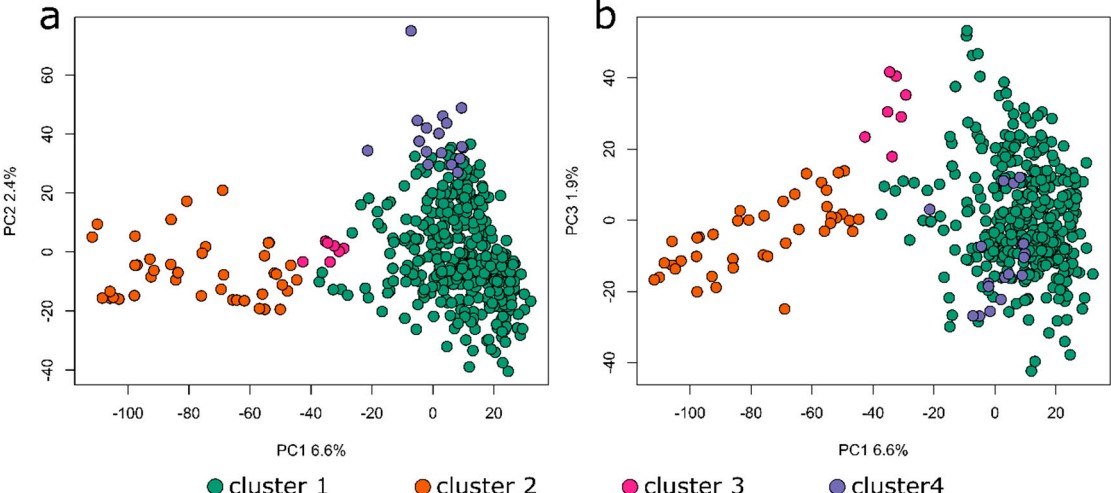

**Figure 5.** Principal component analysis of the population structure as determined from the 1 marker for every 50,000-base, dataset X-Y (**a**) and X-Z (**b**). The marker colors correspond to the populations shown in Figure 4.

## 3.3. Phenotypic Diversity of Numerous Traits of Interest to Breeders

The population was assessed for several different traits under field conditions from plots of plants grown in 2015 (and 2017 in the case of 50% bloom date). The range of phenotypes is shown in Figure 6.

These results are consistent with, but not conclusively indicative, of the hypothesis that this diversity panel will be useful for identifying loci controlling any of these traits.

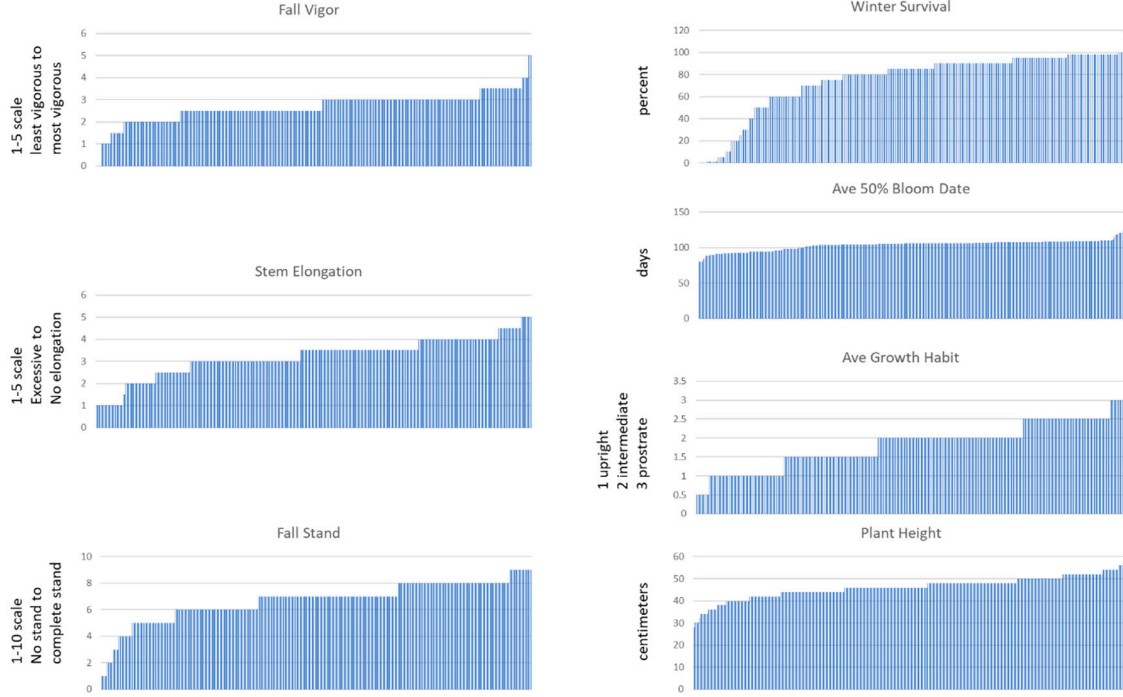

**Figure 6.** Phenotypic variations for several agronomic traits of interest within the diversity panel. Fall vigor, stem elongation, and fall stand were evaluated on a 1:5, 1:5, and 1:10 visual scale, respectively. Winter survival was an estimate percentage, average 50% bloom date was the average days to 50% flowering for 2016 and 2018, average growth habit was scored as 1 = upright, 2 = intermediate, and 3 = prostrate, plant height was the height of the topmost flowers in inches.

## 4. Discussion

Here, we report on a genotyped canola diversity panel, consisting primarily of winter biotypes, that contains sufficient genetic diversity to serve as a valuable resource for identifying loci associated with a multitude of traits. Similar panels have been used effectively in crops such as corn [27] and model plants such as arabidopsis (Arabidopsis thaliana Heyn) [28]. Our diversity panel has been used to identify loci associated with freezing tolerance following deacclimation [14] and, as demonstrated above, contains significant phenotypic diversity for a number of other different traits. Thus, it seems likely that there is a range of genotypic variation within our canola diversity panel controlling traits of significant agronomic importance.

### 4.1. Marker Density and Linkage Decay Are Higher Than Previously Reported

The local linkage decay rate suggests that linkage decay, on average, occurs in less than 23.888kb in our population. However, an analysis of LD decay for individual chromosomes indicated a much smaller interval (as small as 7 kb on Chrm A01, for example) and some regions where the local LD decay interval is very small (<100 bases).

Interestingly, the rate of LD decay rate is significantly greater than has been observed in other diversity panels of canola. For example, LD decay rates for Chrm A01 have been estimated to be as high as ~457 kb in a different population with a marker density of approximately every 29 kb [29]. However, most estimates (for example, [30]) found Chrm A01 to have an LD decay rate of ~30 kb when marker density was at ~1/10 kb. As might be expected, we detected an increase in the LD decay interval when we included fewer rare alleles in our assessment (data not shown). However, inclusion of these rare but not unique alleles (MAF count >5) is justified because they are likely real as they occur

in multiple individuals. These additional alleles could be uncovering more crossover events in the population. The smaller than usual LD decay intervals that we observed could also be due to the fairly high diversity in our population, which could provide a greater number of crossover events. However, given the LD decay rate observed in our population, it is clearly too great in many localized regions to allow for effective association mapping of this population with the Brassica 60K Illumina Infinium SNP array [31]. Fortunately, the SNP marker density from the GBS data appears to be quite sufficient for association mapping since, on average, linked intervals will contain five or more markers.

However, there was significant variation observed among distances contained within R2 values >0.2 such that, for some markers, only a few bases apart could appear to be unlinked. This was reflected in the variation in the local LD decay rate, where there were some fairly large chromosomal regions with virtually no linkage between markers with very small distances between them; see, for example, the region between position 12,835,274 and 14,073,635 on chromosome A01 (Supplemental File S13) where there is essentially no linkage between markers even a few bases apart in this region. Such regions represent chromosomal segments that have been highly randomized in the population and suggest that these are areas of exceptionally high recombination rates. Likewise, there are regions with exceptionally low LD decay rates as shown by the large peaks in Figure 3. Such regions likely represent areas of repressed recombination, such as around centromeric regions with high levels of heterochromatin.

In regions with high linkage decay rates, any association between a given trait and the linked marker would be impossible to identify unless the marker fell directly on the causative mutation. In our parallel study [14], it was rare for a marker to fall into the transcribed region of a gene if the LD decay interval was large. However, when the LD decay interval was very small, the associated marker often fell into the coding region of a gene in over 50% of such cases. However, this information is invaluable for identifying relevant intervals in which to look for candidate genes with high confidence. With the information provided in Supplemental File S13, arbitrary choice of intervals around markers of interest is not needed, and candidate genes can be more easily identified. One other interesting observation is that markers are often clustered along the chromosome. Thus, although the average marker distance is small, the power to detect associations may be somewhat hampered by distances between clusters of markers.

*4.2. Population Structure Provides Information Useful for Genome-Wide Association Studies and Future Breeding Efforts*

The population structure obtained through multiple analysis methods indicated that this diversity panel is highly divergent but consists of two to four primary sub-populations. Thus, the population of canola making up this diversity panel appears to be similarly structured relative to previous populations that have been used in the past for conducting GWAS studies on canola. Several other studies in which similar population structure analyses have been done also identified two to four primary sub-populations [9,13]. The observation that many of the lines contained some proportion of admixtures is likely due to the fact the population has been used primarily for breeding purposes and thus likely contains individuals that have been crossed at multiple times in the recent past. Indeed, this fact might also explain the higher than expected rate of LD decay that we observed since outcrossing might be higher than expected from a primarily self-pollinating species. It is also noteworthy that the majority of the 39 spring lines present in this primarily winter canola diversity panel fall within population two, as shown by the orange color in Figures 4 and 5, with a few spring lines falling within clusters one and three in Figures 4 and 5 (and Supplemental File S7). Additionally, some true winter types also fall within cluster two (Supplemental File S7). This is interesting since there was no obvious segregation of populations based on growth habit in a similar mixed population (see, for example, [32]). It is likely that the clustering that we observed is an artifact resulting from the fact that this is a breeding population rather than a natural population. Thus, some kinship might be expected as the breeding lines for this trait might be selected based on this important agronomic trait. The lack of clustering of spring lines in the study observed [32] was expected since any loss of function

mutation in the FOWERING LOCUS C pathway would result in a spring growth habit and thus might be expected to have arisen multiple times independently. The contamination of this primarily winter rapeseed population by some spring biotypes should be taken into account in any future use of this population to study the genetics of flowering time.

## 5. Conclusions

In conclusion, we have collected and refined a large diversity panel of primarily winter rapeseed lines for GWAS that should be useful in identifying multiple genetic elements controlling numerous traits of agronomic interest. We have made available a well characterized and exceptionally dense set of genetic markers, LD tables, and kinship maps for other researchers interested in using this resource. Finally, we note that this tool has demonstrated usefulness as it has been successfully used to identifying multiple genetic loci associated with freezing tolerance following deacclimation [14] from this population that match those found by other association studies for similar traits in other systems. It is our hope that this resource will be useful for breeders, plant physiologists, and developmental biologists interested in studying various aspects of winter rapeseed biology and diversity.

**Supplementary Materials:** The following are available online at https://doi.org/10.5281/zenodo.4302088. Supplemental File S1: List of ARS accession names with associated common cultivar names and raw data from phenological measures. Supplemental File S2: Original quality-filtered vcf file provided by Cornell University. Supplemental File S3: Imputed VCF file. Supplemental File S4: Imputed VCF file filtered for MAF counts >5. Supplemental File S5: statistical analyses of observed and expected allelic frequency along with Chi square analysis of the results. Supplemental File S6: Kinship matrix. Supplemental File S7: Table showing individual lines associated with each bar on the histograms from Figure 4. Spring lines are highlighted in yellow. Supplemental File S8: Table containing the Nucleotide diversity for each sliding window with the Tajima's D values noted for each window start site. Supplemental File S9: PDF file with image of the unrooted neighbor-joining tree generated from the imputed and filtered VCF file in Tassel5. Supplemental File S10: Linkage disequilibrium decay table. Supplemental File S11: PDF file with images of the regression analysis from the individual A and C genomes. Supplemental File S12: R script used to define local LD decay. This program should work with any LD decay table generated by the program Tassel5. Supplemental File S13: Table providing the start end and mid-point of each window, and the LD decay rate for that window in bases, as well as the model fit statistics. Supplemental File S14: Meteorological data for field study years.

**Author Contributions:** Conceptualization, D.P.H., W.S.C. and J.V.A.; methodology, M.S., J.F. and Z.I.T.; software, M.S., J.F., Z.I.T., A.P.H., and G.A.H.; validation, A.P.H.; formal analysis, all authors; investigation, all authors; resources, J.V.A.; data curation, D.P.H.; writing—original draft preparation, D.P.H.; writing—review and editing, all authors; visualization, all authors supervision, D.P.H.; project administration, D.P.H.; funding acquisition, D.P.H., W.S.C. and J.V.A. All authors have read and agreed to the published version of the manuscript.

**Funding:** This research was funded by The U.S. Department of Agriculture, Agricultural Research Service CRIS# 3060-21220-033-00-D.

**Conflicts of Interest:** The authors declare no conflict of interest.

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
