# Peer review of "A New Diversity Panel for Winter Rapeseed (Brassica napus L.) Genome-Wide Association Studies"

_agronomy, doi:10.3390/agronomy10122006_

Round 1
Reviewer 1 Report
GBS was done for 429 members of canola consisting of winter biotypes. GBS revealed 290 SNPs mapped. Clusters and admixture was identified. LD decay rates were presented in a sliding window highlighting hotspots for recombination.
Comments:
- Imputation is old custom. Imputed set will no more is considered as high quality marker set. Please use SNPs that have 0.05 MAF with above 90% call rate and redo entire analysis
- Imputation could bias admixture, LD decay and recombination hot spots as well as kinship
- Robertson A (1975) - Gene frequency distributions as a test of selective neutrality. Genetics, 81, 775–785; also Robertson A (1975) - Letters to the editor: remarks on the Lewontin- Krakauer test. Genetics, 80, 396.) and Nei & Maruyama (1975) - Nei M, Maruyama T (1975) Letters to the editors: Lewontin-Krakauer test for neutral genes. Genetics, 80, 395 argued that correlated co-ancestry inflates the neutral variance in genetic differentiation when compared to its expectation under an island model of population structure. Perhaps the authors can comment on this?
Author Response
- Imputation is old custom. Imputed set will no more is considered as high quality marker set. Please use SNPs that have 0.05 MAF with above 90% call rate and redo entire analysis.
- Imputation could bias admixture, LD decay and recombination hot spots as well as kinship
We agree that how analyses are done do change over time. Thus, we have included the original vcf file provided to us by Cornell University who did the snp mapping and initial filtering of the GBS data. This way, as things change, anyone who might want to use this population can filter the data as best practices continue to evolve. However, Imputation is still an accepted method with over 8,000 GWAS papers published since 2019 based on a google scholar search. Additionally, we took the added effort to assess the imputation accuracy (which was excellent).
A review of the literature from our best assessment indicated very little impact of imputation on population analysis – as we understand from the review of the literature, this is likely because the kinship/population structure is assessed first by most imputation programs, and that information is used to set the imputed values. Thus, kinship/population structure does not change much between imputed and non-imputed data sets.
Although we disagree that imputation will have a significant effect on population structure and LD decay rates, in our attempt to we discovered that altering the MAF could impact population structure. (for example, see Linck and Battey 2019 “Minor allele frequency thresholds strongly affect population structure inference with genomic data sets”). However, they warm against deleting rare alleles since that appears to result in underestimation of population complexity. They do recommend removing singletons however, and there were about 60K snps in our imputed dataset that were singletons. We have now filtered these out and as per best practices have set the minimum minor allele count to 5 (slightly more stringent than just 2, but far less than 5%) and reran the analyses following this filtration. We appreciate the suggestions as it allowed us identify and this potential issue. Again however, with the raw data provided, if others wanted to alter our snp selection process, they could easily do so.
- Robertson A (1975) - Gene frequency distributions as a test of selective neutrality. Genetics, 81, 775–785; also Robertson A (1975) - Letters to the editor: remarks on the Lewontin- Krakauer test. Genetics, 80, 396.) and Nei & Maruyama (1975) - Nei M, Maruyama T (1975) Letters to the editors: Lewontin-Krakauer test for neutral genes. Genetics, 80, 395 argued that correlated co-ancestry inflates the neutral variance in genetic differentiation when compared to its expectation under an island model of population structure. Perhaps the authors can comment on this?
We are not quite sure what to make of this statement. We are using a breeding population rather than a true natural population. We could see how this might impact the assessment of HW equilibrium state of any given allele. It is generally assumed that the vast majority of snps are neutral and that is indeed the basis for GWAS analysis. This may or may not be the case in our population. However, we are reluctant to comment on this in the manuscript since it is really a bit beyond the point of the manuscript. We did add Tajima’s D and nucleotide variation data to the manuscript which did identify some regions that appear to be under selection. We do hope, however, that we provide the data needed for others to use our data to address such questions and publish their analyses should they so desire, and would be quite happy if our data set was used by others to dig deeper into such questions. Indeed, it is the hope that our dataset would be valuable to others that has motivated us to release this dataset rather than just tacking pieces of it to our own GWAs studies of interest. We hope that the reviewer finds this as an acceptable response.

Reviewer 2 Report
Horvath et al. reported a new winter rapeseed diversity panel for the Brassica research community, which is useful. In this manuscript, the authors used genotyping-by-sequencing to check the genetic diversity and population structure of the studied lines. They also released some phenotypic data collected by them, which are valuable. However, before the acceptance of this manuscript, I do have some questions which may need addressing:
Major:
- Line 102-104, Page 3: Since some lines might be misclassified (not winter type), why did authors retain these lines in the analysis and show them in the result? Would these lines confuse the analysis?
- Line 210-213, Page 4: When collecting the phenotypic data, how did the authors avoid artificial evaluations or evaluation biases? Will different evolutions lead to different findings in the downstream analysis (e.g. GWAS)?
- There are several findings or conclusions cited in this manuscript which comes from another research by the team. However, the research is not published but submitted. Can the authors find other relevant publications to cite instead of using their parallel one?
- Line 255, Page 7: Can the authors show a phylogenetic cluster of the population?
Minor:
- Delete line 37-43 on page 1
- Line 166, Page 4: Please justify the selection for “centered IBS” or add a reference here
- Line 229, Page 6: What methods did the authors use to solve the heterozygous genotype imputation problem?
- Line 232, Page 6: To my knowledge, Hardy-Weinberg equilibrium is more suitable for diploid. Is this suitable for polyploids? Maybe delete the information here
- In the SNP section, the authors may add nucleotide diversity (π) and Tajima’s D analyses
- The resolution of figures is low. Maybe increase it.
Author Response
Major:
- Line 102-104, Page 3: Since some lines might be misclassified (not winter type), why did authors retain these lines in the analysis and show them in the result? Would these lines confuse the analysis?
This should not confuse the analysis. There have been several studies with mixed populations of winter, spring, and semi-winter types that have shown that these growth habits do not influence population structure- see for example Wrucke et al. 2020 “Genome‐wide association study for frost tolerance in rapeseed/canola (Brassica napus) under simulating freezing conditions” Plant breeding 139 356-367. Indeed, since these growth habits can arise by multiple loss of function mutations in the FLC flowering pathway, it is highly likely that this trait has spontaneously occurred multiple times in many different lineages. It might however impact any GWAs designed to assess flowering time, and thus we have made certain to not which lines are spring types so that they can be included or excluded as needed by researchers interested in this trait. We now note this specific possibility in the manuscript. See “Although all of the lines were presumably winter biotypes, 19 lines may have been mis-classified based on the observations that they flowered without the need for vernalization, and 20 lines were known spring types. These lines should be considered for any projects to map flowering time genes. Of the primarily winter rapeseed lines, 238 had been advanced to the F9 generation via selfing prior to DNA extraction. The remaining lines had been advanced to between the F5 and F8 generation. A list of common cultivar names associated with the ARS accession numbers and preliminary phenotypic traits including two years of data for flowering time are available in supplemental file 1.” Also see the last paragraph in the discussion. Additionally, all spring lines can easily be identified in both supplemental file 1 and 7. Thus, anyone wishing to use this population for other studies can easily identify them and include or exclude them as they see fit.
- Line 210-213, Page 4: When collecting the phenotypic data, how did the authors avoid artificial evaluations or evaluation biases? Will different evolutions lead to different findings in the downstream analysis (e.g. GWAS)?
The phenotypic traits shown are merely there to indicate potential variability in traits of agronomic interest. We do not try to make any claim that these data are useful except to show the ”potential” of our p[population for identifying genes of agronomic interest. Please note the sentence in the results “These results are consistent with, but not conclusively indicative, of the hypothesis that this diversity panel will be useful for identifying loci controlling any of these traits.”
- There are several findings or conclusions cited in this manuscript which comes from another research by the team. However, the research is not published but submitted. Can the authors find other relevant publications to cite instead of using their parallel one?
These citations are just there to show that this particular set of snps can be used to in a GWAS to identify probable genes of interest in this population. The Horvath et al paper 2020, was just accepted and the Chao et al paper was initially rejected from The Plant Genome because much of the data presented in this manuscript was missing from it. The Chao et al. paper will be submitted to Agronomy as soon as this manuscript is accepted. We anticipate no significant issues with the GWAS for freezing tolerance study presented in the Chao et al paper. If the Chao et al paper has not been accepted before this paper has passed through the copy editor, we can remove that reference. However, as one is likely sufficient to show that the population is functional we have deleted the reference to the Chao paper. That said, this population has not previously been genotyped and thus was not available for any other GWAS, and thus these are the only two studies using this population.
- Line 255, Page 7: Can the authors show a phylogenetic cluster of the population?
LOL! Yes, we have now included it in this version of the manuscript as a supplemental file. We had this as a figure in a previous version but one of the previous reviewers thought it was not needed so we removed it.
Minor:
- Delete line 37-43 on page 1
Fixed
- Line 166, Page 4: Please justify the selection for “centered IBS” or add a reference here
It was selected because that is the default option in Tassel5. If the reviewer knows that it was an inappropriate choice, please advise. We would be happy to change the method if a better one exists.
- Line 229, Page 6: What methods did the authors use to solve the heterozygous genotype imputation problem?
We ignored this for the most part. As the reviewer is aware, few imputation programs impute heterozygotes, and NGSEP is no exception. We were not too worried about this as an issue because we had very little heterozygosity as would be expected from a primarily self pollinating species such as B. napus. We added some text to that effect.
- Line 232, Page 6: To my knowledge, Hardy-Weinberg equilibrium is more suitable for diploid. Is this suitable for polyploids? Maybe delete the information here
The line was deleted. However, we are talking about unique loci and one might expect the majority of them to be neutral and thus be maintained at HWE in a natural population. The fact that so few were observed was a surprising observation -even in a breeding population that might be expected to be undergoing significant selective pressures.
- In the SNP section, the authors may add nucleotide diversity (π) and Tajima’s D analyses
We have added this information to the manuscript. There are some interesting observations where there is a very localized dip in Pi that corresponds to a very negative Tajima’s D value that could be located close to agronomically important genes under selection as this is a breeding population. Hopefully this manuscript will be useful to population geneticists interested in investigating such things using the data we present here.
- The resolution of figures is low. Maybe increase it.
We will leave that up to the copy editor. Currently the figures are scalable since they were mostly created in excel.

Round 2
Reviewer 2 Report
Thanks the authors for addressing my previous concerns and I have no more questions. The manuscript could be accepted.